# Investigation of OFDM-Based HS-PON Using Front-End LiFiSystem for 5G Networks

Meet Kumari [1], Mai Banawan [2], Vivek Arya [3] and Satyendra Kumar Mishra [4],*

1   Department of Electronics and Communication Engineering, UIE, Chandigarh University, Gharuan, Mohali 140413, India; meetkumari08@yahoo.in
2   Electrical Engineering Department, Alexandria University, Alexandria 21544, Egypt; mai.a.f.banawan@alexu.edu.eg
3   Department of Electronics and Communications Engineering, FET, Gurukula Kangri (Deemed to Be University), Haridwar 249404, India; ichvivekmalik@gmail.com
4   SRCOM, Centre Technologic de Telecomunicacions de Catalunya, Castelldefels, 08860 Barcelona, Spain
*   Correspondence: smishra@cttc.es

**Abstract:** Fifth-generation (5G) technology has enabled faster communication speeds, lower latency, a broader range of coverage, and greater capacity. This research aims to introduce a bidirectional high-speed passive optical network (HS-PON) for 5G applications and services including mobile computing, cloud computing, and fiber wireless convergence. Using 16-ary quadrature amplitude modulation orthogonal frequency division multiplexing techniques, the system transmits uplinks and downlinks with a pair of four wavelengths each. Light fidelity (LiFi) services are provided with blue light-emitting-diode-based technology. With a threshold bit error rate (BER) of $10^{-3}$, the results demonstrate reliable transportation over a 100 km fiber at $-17$ dBm received power and in a maximum LiFi range of 20 m. Furthermore, the system offers symmetric $4 \times 50$ Gbps transmission rates under the impact of fiber–LiFi channel impairments with maximum irradiance and incidence half-angles of 500. Additionally, at threshold BER, the system provides a detection surface range from 1.5 to 4 cm$^2$. Compared to existing networks, the system also provides a high gain and low noise figure. A number of features make this system an attractive option. These include its high speed, high reach, high split ratio, low cost, easy upgradeability, pay-as-you-grow properties, high reliability, and ability to accommodate a large number of users.

**Keywords:** 5G; HS-PON; LiFi; OFDM; PON; TWDM





## 1. Introduction

Several countries around the world are currently deploying fifth-generation (5G) mobile networks. In comparison with 4G, 5G relies on key enabling technologies such as software-defined networking, massive multi-input–multi-output, network function virtualization, big data, edge computing, etc. In several industries, such as automotive, healthcare, media, transportation, smart cities, and entertainment, 5G enables bandwidth-demanding applications such as virtual reality, augmented reality, industrial automation, automated driving, and high-definition video streaming. About 70% of the traffic generated via these applications comes from indoor users. As a result of the limited spectrum, radio frequency (RF)-based systems will face challenges in meeting the high demand for 5G and beyond 5G networks. In the next generation of indoor optical wireless communication (OWC), visible light communication (VLC) offers a promising solution for reducing infrastructure requirements [1].

As an advanced OWC technique, light fidelity (LiFi) overcomes limitations in RF-based systems [2]. The LiFi system relies on light emitting diodes (LEDs), which are cost-effective and energy-efficient. LiFi technology is safe for the human body since it reduces exposure to RF electromagnetic fields. It is widely applicable, such as in offices, homes, conference

rooms, subways, etc. LiFi technology provides high transmission rates using the unlicensed visible frequency range (400–800 THz), which is ten thousand times wider than the entire RF spectrum. Since it cannot penetrate objects, it provides a secure transmission method [3,4]. The characteristics of LiFi make it an ideal candidate for 5G technology. However, pure LiFi networks have inherent shortcomings, such as link vulnerability and limited coverage, which negatively affect the quality of service to users [5]. In areas with limited RF wireless and wired communications, LiFi can be integrated to optical fibers for indoor networks. Hybrid LiFi–fiber links offer high transmission rates, wide coverage ranges, and mobility for 5G mobile networks [6].

The passive optical network (PON) has become the primary counterpart of 5G fron-thaul and backhaul networks [7]. PON technology has evolved through three generations, 1 Gbps (2000–2005), 10 Gbps (2010–2015), and 25/50 Gbps (2020). By 2030, PONs will provide 100 and 200 Gbps per wavelength due to improved bandwidth efficiency. It is expected that a consensus optical access network in 5G/B5G will be required not only for urbanization, but also for mobile X-haul and business services [8]. There has been a significant increase in vendor interest in time and wavelength division multiplexed (TWDM) PON technology. A shared optical distribution network (ODN) supports multiple optical line terminals (OLTs) with the TWDM-PON [9]. The TWDM-PON is capable of multi-wavelength operation as well as wavelength tunability, enabling enhanced network functionality not possible with prior generations of the PON. Moreover, it offers 'pay-as-you-grow' deployment of OLT transceiver units and a smooth upgrade of the aggregated capacity in the ODN [10]. The TWDM-PON has the advantage of sustaining high transmission rates with simpler and less complex transceivers [11]. There is a need to improve the transmission data rate for the 25 G TDM-PON to meet the bandwidth demand caused by new "hungry" applications. International Telecommunication Union-Telecommunication Standardization (ITU-T) and Institute of Electrical and Electronics Engineers (IEEE) continue to work on standardizing the 25 G TDM-PON. In anticipation of the future 50 G-PON or high-speed PON (HSP-PON), they have been seeking optimal modulation formats and technologies [12].

OFDM is considered to be a promising multiplexing technique for HSP-PONs due to its high spectral efficiency, fiber dispersion resistance, and dynamic resource allocation [13]. LiFi is therefore able to enhance the tolerance to fiber nonlinearities, large fluctuations, and noise introduced by LEDs in a hybrid HS-PONs/OFDM system [14].

The work investigates new network architectures in order to satisfy the data traffic demand, which is continuously increasing. It uses visible light for transmission as well as luminance and is high-speed, bidirectional, and fully optically networked [15]. Aside from providing excellent benefits, it offers low costs, increased data security, and many more. LiFi systems can be made flexible, adaptable, and simple by using OFDM multiplexing techniques. Equipment is connected separately to several orthogonal subcarriers in such systems. A key advantage of OFDM is that it effectively mitigates inter-symbol interference caused by light reflection, allowing multiple access and power loading to be adjusted on each subcarrier [16]. A hybrid HS-PON/OFDM with a LiFi system provides the ability to transmit data over high-bandwidth fiber–LiFi links ("50 Gbps" per channel) [17].

1.  This paper proposes and investigates the performance of an HS-TWDM PON/OFDM system that utilizes LiFi. Data transmission over fiber–LiFi links based on 5G is addressed using our proposed system [18], serving as the front-end communication network for deployed broadband access networks. The effects of fiber attenuation, linear and non-linear impairments, dispersion, noise in LiFi channels, and misalignment are considered.
2.  We analyze the system's performance in terms of the bit error rate (BER), received spectra, receiver sensitivity, and optical signal to noise ratio (OSNR).
3.  We compare our results with those of prior established studies in order to verify the results we obtained.

An overview of the proposed model is presented in Section 2 with a conceptual diagram and mathematical description. As well as illustrating the obtained results, Section 3 analyzes the proposed model. The conclusion is presented in Section 4 along with future prospects.

## 2. Proposed Model

Figure 1 depicts the conceptual diagram of the PON/LiFi model for a 5G network.

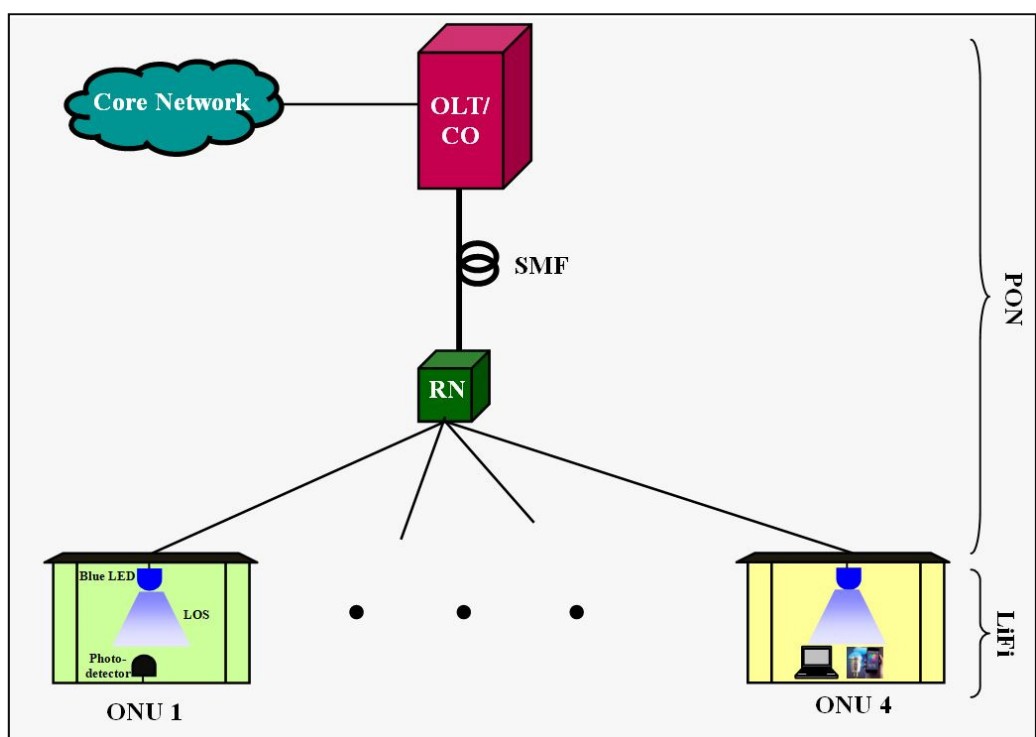

**Figure 1.** Conceptual diagram of the proposed PON/LiFi model.

Signals transmitted from the core network's OLT are distributed to four optical network units (ONUs) via a single-mode fiber (SMF) in an optical distribution network (ODN). The PON integrates a line-of-sight (LOS) indoor optical wireless channel at the front end to provide a high-quality signal to each ONU. An LED in a room serves users' equipment such as computers, phones, and other smart devices in indoor LiFi scenarios. The optical wireless network in the visible spectrum constitutes a blue LED (450 nm or 666.2 THz; bandwidth = 1 GHz). The LED receives data from the PON via an SMF and stores it. The light flashes at an extremely high speed that is invisible to the human eye. At the receiver, a photodetector (PD) reads all scintillations to extract the data [19].

It employs OFDM modulation with four downlink (DN) frequencies (187.5, 187.6, 187.7, and 187.8 THz) and four uplink (UP) frequencies (195.3, 195.4, 195.5, and 195.6 THz), following the ITU-T channel spacing of 100 GHz. Due to its higher signal to noise ratio (SNR) and better receiver sensitivity, 16-QAM modulation is used for OFDM modulation. In order to support high-speed and long-haul transmissions, coherent detection is used to take into account fiber impairments and LiFi channel noise [20]. For 5G-based hybrid wired–wireless transmission, Figure 2 shows a systematic block diagram of a full-duplex HS-PON/LiFi model using 16-QAM-OFDM modulation over the backend SMF link and the frontend LiFi channel link.

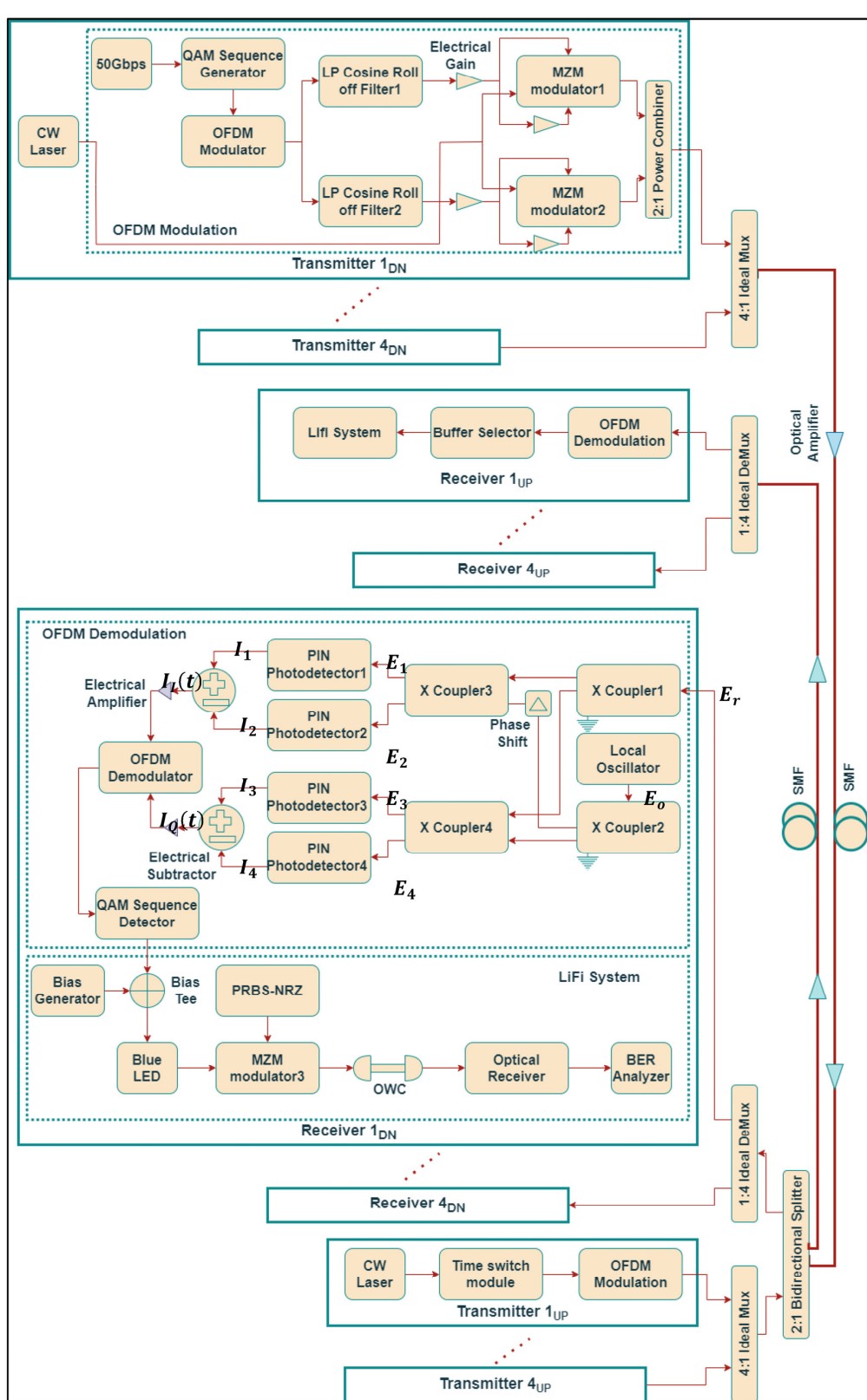

**Figure 2.** Block diagram of proposed OFDM-based bidirectional HS-PON using LiFi model.

OptiSystem software was used to design and analyze this model. Simulation parameters include 32,768 samples and 50×109 symbols/sec. There are four transmitters and receivers used for both DN and UP transmission signals, as shown in the figure. DN transmitters (TX) use continuous-wave laser diodes with a −6 dBm CW output power and 0.15

MHz linewidth. Downlink transmitters generate a random bit sequence at a high bit rate of 50 Gbps that is fed into a 16-QAM sequence generator that encodes four bits per symbol. A 16-QAM signal is modulated using an OFDM modulator with 1024 fast Fourier transform points, 512 subcarriers, and 64 cyclic prefix points. Two low-pass cosine filters (with a 0.2 roll off factor) are used to filter the in-phase (I) and quadrature (Q) signals generated by the OFDM modulator, followed by two electrical gain components (gain = −0.008). Mach–Zehnder modulators (MZMs) then process the signal. In order to combine the generated modulated signals at a specific frequency, a 2:1 power combiner is incorporated. Thus, four HS-PON/OFDM downlink signals are multiplexed via a 4:1 ideal Mux at four downlink frequencies between 187.5 and 187.8 THz. SMFs are used to transmit signals considering fiber attenuation, nonlinearities, and dispersion. Pre- and post-amplification over the SMF link was achieved with two optical amplifiers with gains of 13 dB. Using a 2:1 bidirectional splitter, downlink signals can be directed to a specific LiFi user at the downlink receiver. In order to split the downlink signals into two pairs, 50:50 couplers are used. Coherent orthogonal reception at ONU is enabled by this setup, as well as a local oscillator and 900-phase shift device. The signal is converted from the optical domain into electrical domain using two pairs of PIN PDs, followed by two electrical subtractors, followed by two electrical amplifiers. The QAM sequence detector and OFDM demodulator are then used to demodulate the signal. The QAM sequence detector and OFDM demodulator use the same parameters as those in the modulation process at the OLT to reconstruct transmitted signals. To determine the binary sequences as well as the mapped electrical signals, a QAM sequence detector is used [20,21].

In the LiFi system, the direct current bias signal with a voltage of 4 V is integrated with the electrical signals from the QAM detector to drive a blue LED (450 nm). LEDs operate in linear regions with 65% quantum efficiency. The incoming LED input is modulated with pseudo-random bit sequences in non-return to a zero format and then forwarded to the LiFi communication channel by an MZM. We also consider interference from external light sources and channel noise. To capture the transmitted signal, an optical receiver is used. In addition, BER analyzers are used to analyze the received information in terms of BER and eye patterns [22].

For uplink data transmission, 4 × 150 Gbps OFDM signals operating at 195.3–195.6 THz frequencies are generated at the ONU and detected at the OLT using OFDM demodulation. In the simulation process, a buffer selector is included to select the last iteration. Figure 3 illustrates the time switch section, which consists of seven time slot modules. Each time slot module incorporates two cascaded 2 × 1 dynamic optical switches (dynamic Y select blocks) for transmission at a specific timer slot. In Figure 4, we introduce the proposed model's LOS propagation scenario. To isolate uplink and downlink transmission and reduce interference between the channels, two separate fiber links are used in this model. For high signal quality and coverage across large areas, LiFi communication systems use LOS transmission.

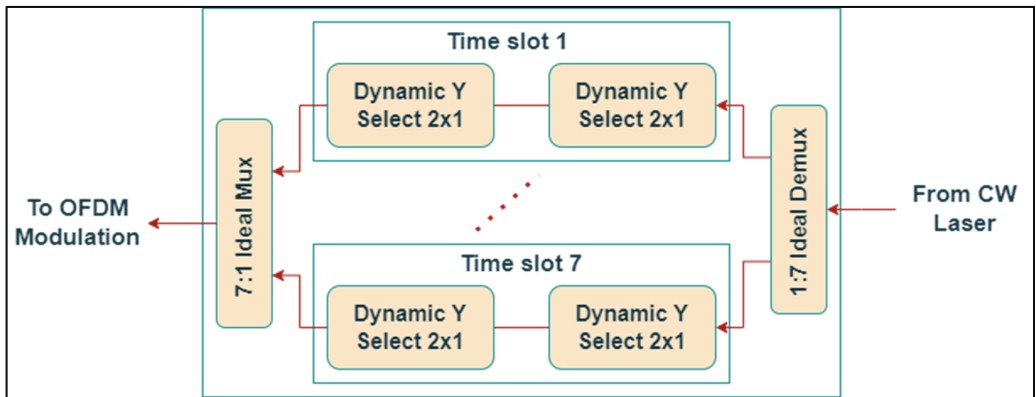

**Figure 3.** Block diagram of time switch module for uplink data transmission in the proposed model.

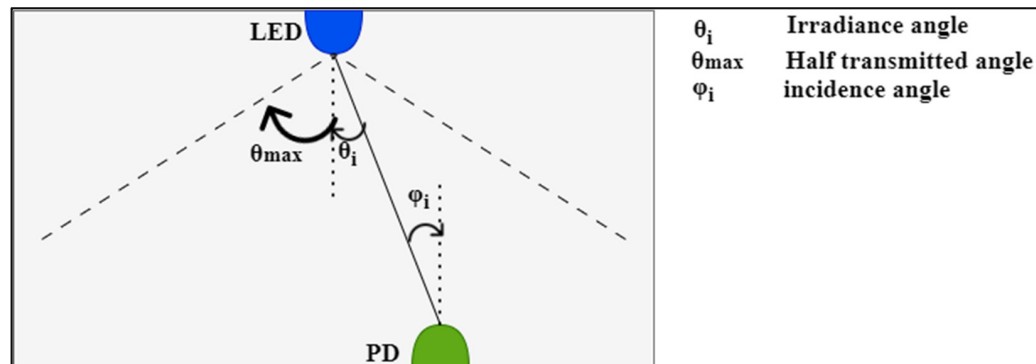

**Figure 4.** LiFi-LOS propagation scenario in the proposed model.

*Mathematical Description*

In each TX, the generated optical signal from the laser diode is passed through the OFDM modulator having a 16-QAM modulation format. For a low-pass cosine roll off filter, the transfer function is given as follows [23]:

$$X(f) = \begin{cases} \sigma & (|f| < f_1) \\ \sqrt{0.5\sigma^2 \left[1 + cos\left(\pi.\frac{|f|-f_1}{r_a \Delta f_{FWHM}}\right)\right]} & (f_1 \le |f| \le f_2) \\ 0 & (f_2 \le |f|) \end{cases} \tag{1}$$

where $\sigma$ is the insertion loss, and $r_a$ is the roll off factor parameter. $\Delta f_{FWHM}$ is the full width at a half-maximum frequency between $f_1$ and $f_2$. Here, for the cutoff frequency, $f_c$, parameters $f_1$ and $f_2$ are given as follows [23]:

$$f_1 = 1 - r_a f_c (0 \le r_a \le 1) \tag{2}$$

and

$$f_2 = 1 + r_a f_c (0 \le r_a \le 1) \tag{3}$$

The received I and Q component signals are phase-shifted at 90° using a 90° phase shift at the coherent receiver, as shown in Figure 2. Mathematically, the optical hybrid outputs disregarding imbalance/loss can be given as follows [24]:

$$E_1 = 0.707[E_r + E_o] \tag{4}$$

$$E_2 = 0.707[E_r - E_o] \tag{5}$$

$$E_3 = 0.707[E_r - jE_o] \tag{6}$$

$$E_4 = 0.707[E_r + jE_o] \tag{7}$$

where $E_r$ is the received signal and $E_o$ is the local oscillator input at RX. For two PDs, the generated photocurrents $I_1$, $I_2$, $I_3$, and $I_4$ are given as follows [24]:

$$I_1 = 0.5\left\{|E_r|^2 + |E_o|^2 + 2Re[E_r E_o^*]\right\} \tag{8}$$

$$I_2 = 0.5\left\{|E_r|^2 + |E_o|^2 - 2Re[E_r E_o^*]\right\} \tag{9}$$

$$I_3 = 0.5\left\{|E_r|^2 + |E_o|^2 + j[E_r E_o^*]\right\} \tag{10}$$

and

$$I_4 = 0.5\left\{|E_r|^2 + |E_o|^2 - j[E_r E_o^*]\right\} \tag{11}$$

Then, the detected signal comprising in-phase and quadrature-phase components is given as follows [24]:

$$I(t) = I_I(t) + jI_Q(t) = 2E_r E_o^* \tag{12}$$

Again, the photocurrent noise contribution (shot-, thermal-, and phase intensity-induced noise) in the total photocurrent is expressed as follows [25]:

$$\left\langle i_{totalnoise}^2 \right\rangle = \left\langle i_{shot}^2 \right\rangle + \left\langle i_{thermal}^2 \right\rangle + \left\langle i_{PIIN}^2 \right\rangle = 2eBI + \frac{4KTB}{R_L} + BI^2 \tau_c \tag{13}$$

where $e$ is the electronic charge, $B$ is the filter bandwidth, $I$ is the PD current, $K$ is the Boltzmann constant, $T$ is the absolute temperature, $R_L$ is the load resistance, and $\tau_c$ is the optical source coherent time. Further, the BER for the proposed model using m-QAM OFDM modulation is evaluated as follows [22]:

$$BER_{m-QAM} = \frac{2}{log_2(m)} \left( 1 - \frac{1}{\sqrt{m}} \right) . erfc \left( \sqrt{\frac{3log_2(m)}{2(m-1)}} \times SNR \right) \tag{14}$$

where $erfc(.)$ represents the complementary error function, and $m$ is the QAM modulation order. Here, $SNR$ is given as follows [22]:

$$SNR = P_T \left( \sqrt{\left( \frac{2hfB}{\eta} \right) P_T + \left( \frac{hf}{\eta e} \right)^2 \left( \frac{4KTB}{R_L} \right)} \right)^{-1} \tag{15}$$

where $P_T$ is the average transmittance power, $h$ is Plank's constant, $f$ is the operating frequency, and $\eta$ is the PD quantum efficiency.

Furthermore, the LOS frequency response LiFi model is given as follows [26]:

$$F_{LOS}(f) = (1 - \vartheta) \frac{n+1}{2\pi s^2} X cos^n (\theta) cos(\mu) rect \left( \frac{\mu}{\varphi} \right) \tag{16}$$

where $X$, $\theta$, and $\mu$ indicate the detector area, TX radiance angle as well as RX incidence angle, respectively. Here, $n$ indicates the Lambertian emission order. $\vartheta$ means represents the blockage status. The symbol $s$ presents the distance between the access point and user equipment. Also, $rect \left( \frac{\mu}{\varphi} \right) = 1$ for $0 \leq \mu \leq \varphi$ and 0 otherwise. $\varphi$ means represent the field of view.

Moreover, the LOS LiFi channel using an LED source is represented as follows [27]:

$$p(t) = q(t) * h(t) + n(t) \tag{17}$$

where $p(t)$ means represents the received signal, $q(t)$ is a transmitted signal, $h(t)$ is an impulse response of the LiFi channel, $n(t)$ is the noise signal, especially Gaussian signal-independent noise (ambient light, thermal noise, and shot noise), and * signifies the convolution operation.

Additionally, the optical signal received $(P_R)$ in the LiFi model is described as follows [27]:

$$P_R = P_T \mu_T \mu_R L_p(\lambda, l) \left( \lambda, \frac{D}{cos\alpha} \right) \frac{A cos\alpha}{2\pi D^2 (1 - cos\alpha_0)} \tag{18}$$

where $\mu_T$ and $\mu_R$ mean represent the optical efficiency at the transmitter and receiver, respectively, $L_p$ means represents the propagation loss factor considering the wavelength, $\lambda$, and length, $l$. Also, $D$ is the upright distance in the transmitter as well as in the receiver, $\alpha$ means is the angle between the vertical receiver plane and the transmitter–receiver trajectory, $\alpha_0$ is the laser beam divergence angle, and $A$ means is the receiver aperture area.

The expression of BER in the LiFi system considering the direct detection OOK modulation technique in is given as follows [27]:

$$BER = 0.5 erfc\left[\frac{n_1 T - n_0 T\left\{(n_1 T)^{1/2} + (n_2 T)^{1/2}\right\}^{-1}}{\sqrt{2}}\right] \tag{19}$$

where $n_1$ and $n_0$ define additive noise sources owing to dark counts as well as background illumination considerably. Besides this, the evaluated performance metrics of the proposed model include the gain, noise figure, SNR, and OSNR and expressed as [28] follows:

$$Gain = 10 log_{10}\left(\frac{Output\ Power}{Input\ Power}\right) dB \tag{20}$$

$$Noise\ Figure = 10 log_{10}\left(\frac{SNR_i}{SNR_o}\right) dB \tag{21}$$

$$OSNR = 10 log_{10}\left(\frac{Signal\ Power}{Noise\ Power}\right) dB \tag{22}$$

Table 1 depicts various simulation parameters used in the design.

**Table 1.** Parameters values [27].

| Parameters | | Values | Unit |
|---|---|---|---|
| CW Laser | Frequency (DN) | 187.5,187.6, 187.7, 187.8 | THz |
| | Frequency (UP) | 195.3,195.4, 195.5, 195.6 | |
| | Input power (DN and UP) | −6 | dBm |
| | Linewidth | 0.15 | MHz |
| LiNb Mach–Zehnder Modulator | Extinction ratio | 60 | dB |
| Multiplexer | Bandwidth | 15 | GHz |
| Optical amplifier | Gain | 13 | dB |
| | Noise figure | 4 | |
| Optical fiber | Length | 10–100 | Km |
| | Attenuation | 0.2 | dB/km |
| | Reference wavelength | 1550 | nm |
| | Dispersion | 16 | ps/nm/km |
| | Dispersion slope | 0.075 | ps/nm$^2$/km |
| | Third-order dispersion | Yes | |
| | Group velocity dispersion | Yes | |
| Photodetector PIN | Responsivity | 1 | A/W |
| | Dark current | 10 | nA |
| | Thermal noise | $100 \times 10^{-24}$ | W/Hz |
| | Shot noise | Yes | |
| LED | Blue | 450 | nm |
| | Quantum efficiency | 0.65 | |
| LOS channel | Distance | 10–20 | m |
| | Room size | $10 \times 10 \times 10$ | m |
| | Transmitter half-angle | 60 | deg |
| | Irradiance half-angle | 20–50 | |
| | Incidence half-angle | | |
| | Detection surface area | 1.5–4 | cm$^2$ |
| Low-pass filter | Cutoff frequency | $37.5 \times 10^9$ | Hz |

## 3. Results and Discussions

Our proposed LiFi HS-PON/16-QAM OFDM system was simulated using OptiSystem v.21. OptiSystem is an integrated software design suite which enables consumers to plan,

test, as well as simulate optical links in the transmission layer of contemporary optical networks. The proposed model is analyzed considering practical scenario under the impact of real environment conditions to deploy a practical design. In the proposed system both mathematical and the OptiSystem simulation analysis have been performed to provide the realistic values of the received outputs in terms of SNR, OSNR, BER, output power, etc.

We simulated four downlink and four uplink channels operating at 187.5–187.8 THz and 195.3–195.6 THz with a 50 Gbps/channel data rate. For 5G networks, it is reported that blue LEDs are used for indoor LiFi LOS transmission. Our simulation includes both linear (e.g., optical noise, crosstalk, and fiber dispersion) and nonlinear impairments (e.g., four-wave mixing). We investigate the model's performance with respect to BER, received power, constellation diagrams, eye patterns, and optical spectra. To analyze performance, we consider different fiber lengths, system throughput, the detection surface area for LiFi channels, transmitter and receiver angles, and the indoor wireless range.

In the proposed work, a minimum BER of $10 \times 10^{-3}$ for LiFi systems is considered sufficient. The proposed digital coherent receivers are implemented with, most likely, forward error correction (FEC) coded sequences. FEC codes allow the recovery of considerable error bit levels to deliver an absolute digital signal over long distances and at high traffic rates. In addition, FEC enhancement utilizes decision algorithms (soft or hard) to allow an optimum net coding gain, better tolerating transmission link impairments and increasing transmission capacity (>100 Gbps) while considering a 7–35% overhead [29,30].

Figure 5a–p depict the attained output plot foreach stage of the hybrid HSP-PON/ OFDM with LiFion50 km fiber with a 10 m LiFi channel range at a 50 Gbps/channel rate for the downlink OFDM signal (187.7 THz).

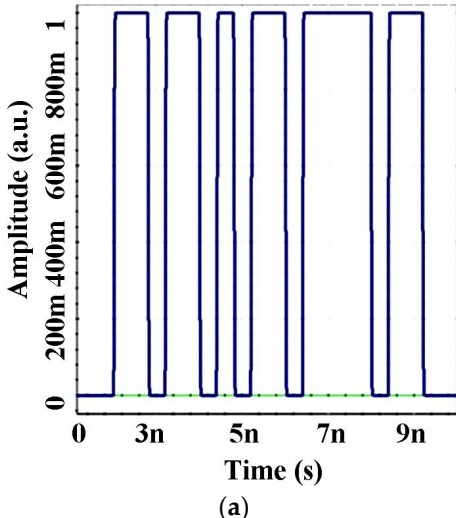

(**a**)

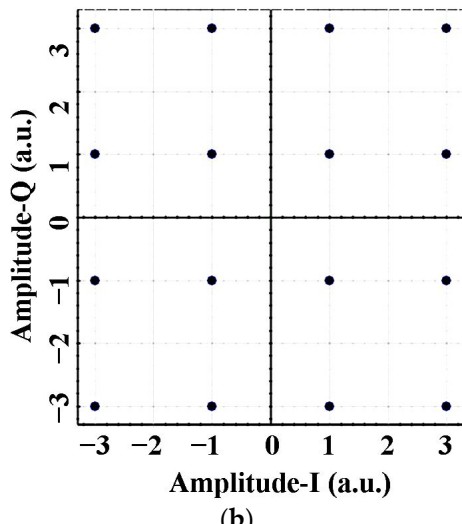

(**b**)

**Figure 5.** *Cont.*

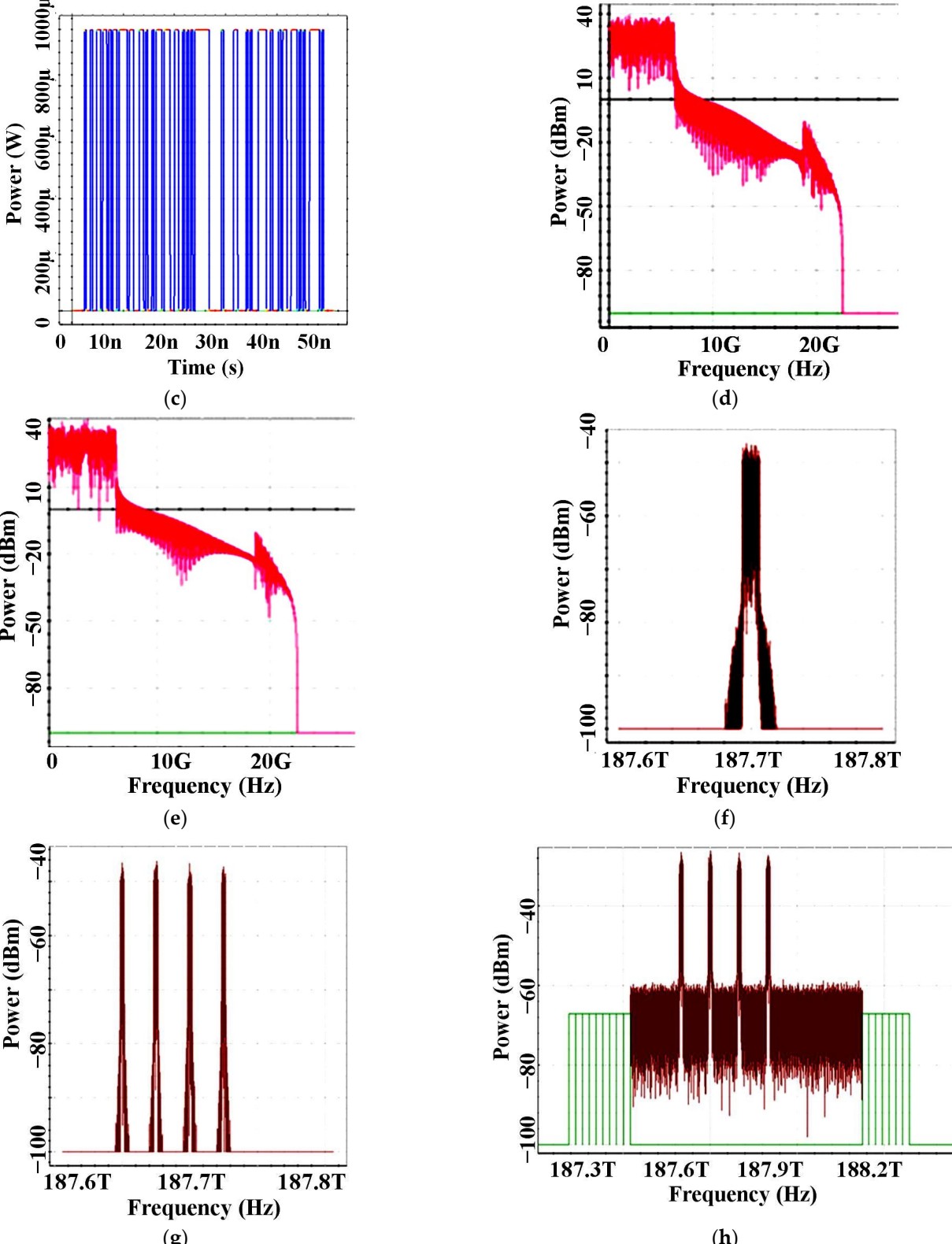

**Figure 5.** *Cont*.

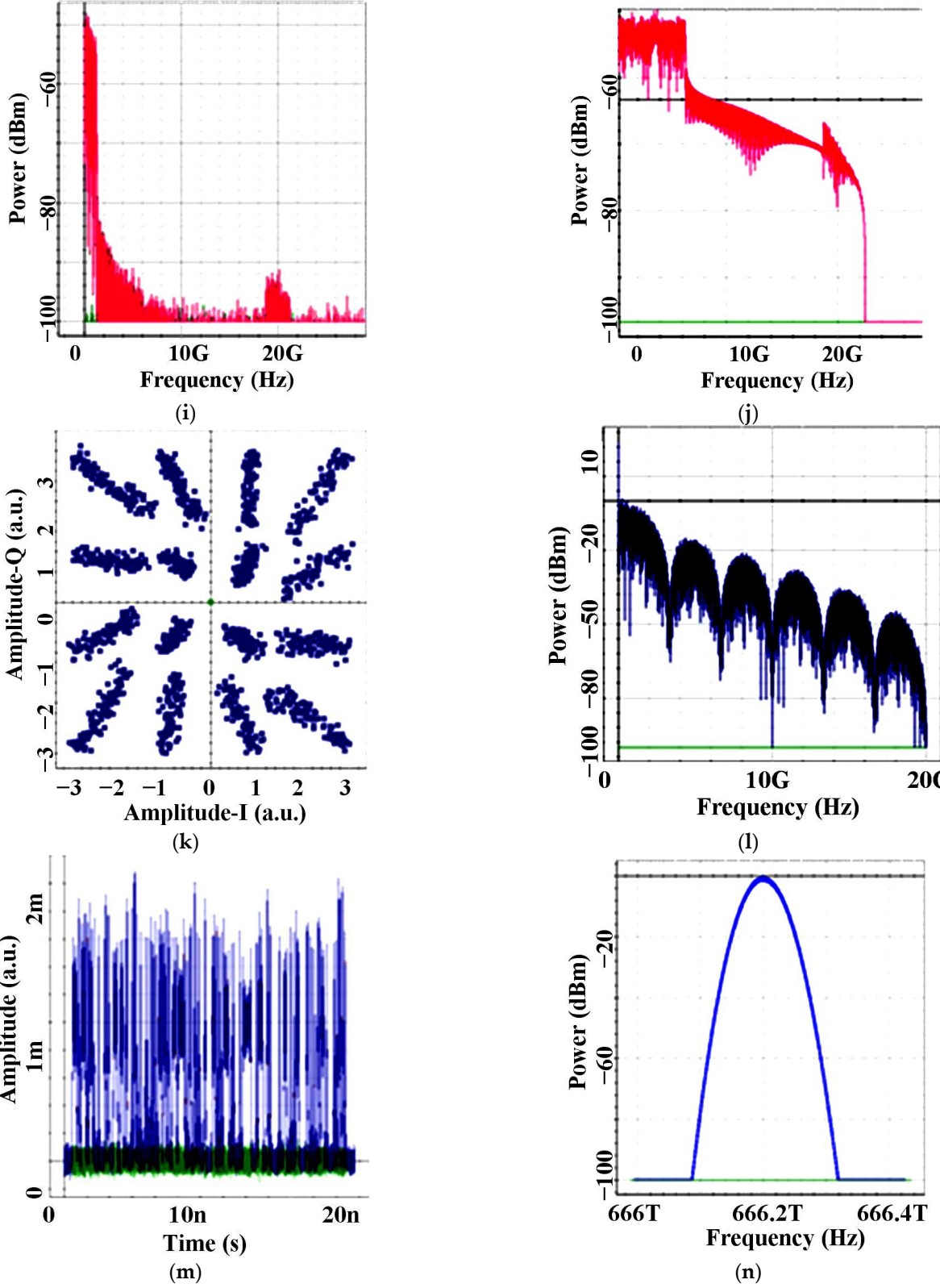

**Figure 5.** *Cont.*

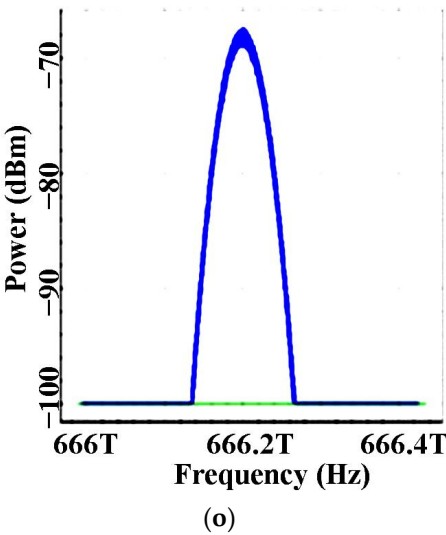

(o)

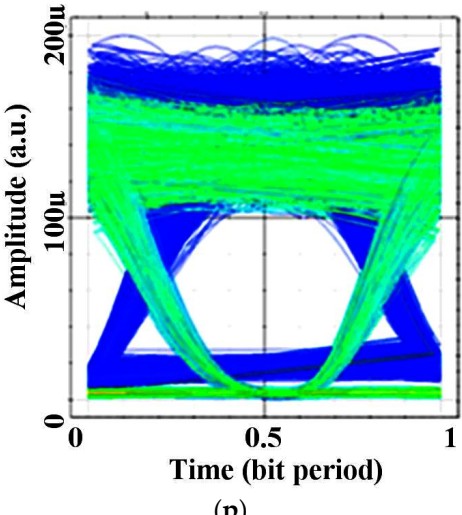

(p)

**Figure 5.** (**a**) Bit sequence at 50 Gbps; (**b**) 16-QAM constellation diagram at Tx; (**c**) timing diagram at Tx; LP cosine filter output for (**d**) I-channel, (**e**) Q-channel, and OFDM modulation signal at Tx; emission spectra of modulation outputs (**f**) single channel transmitted signal at 187.7THz (**g**) before SMF and (**h**) after 50 km SMF; demodulated output for (**i**) I channel and (**j**) Q channel; (**k**) 16-QAM constellation diagram at Rx; (**l**) electrical spectrum at QAM sequence decoder; (**m**) timing diagram at Rx; (**n**) optical spectrum of blue LED; (**o**) optical spectrum after 10 m LiFi range; (**p**) eye pattern in BER analyzer.

Figure 6a,b indicate the optical spectra before the transmission of the proposed work for both downlink (187.5, 187.6, 187.7, and 187.8 THz) and uplink (195.3, 195.4, 195.5, and 195.6 THz) transmission channels with a 100 GHz channel spacing.

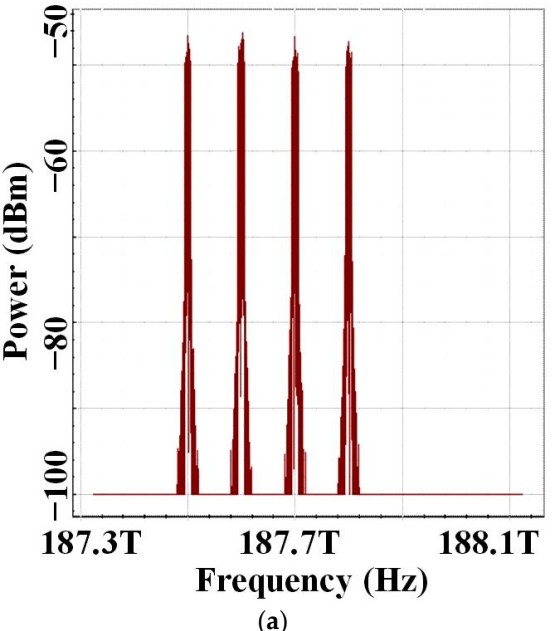

(a)

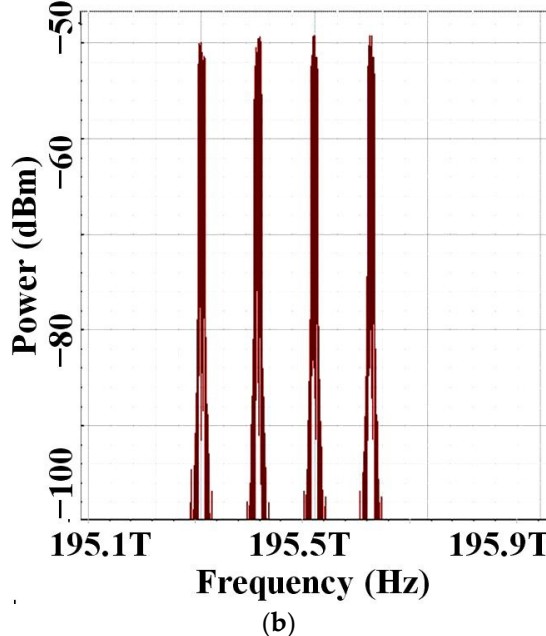

(b)

**Figure 6.** Optical spectra of (**a**) DN and (**b**) UP transmission.

Figure 7a,b represent the received optical power of different frequency signals with a fixed LiFi channel range of 10 m for downlink and uplink directions. As the SMF length increases, the received power decreases for all transmitted signals. In the proposed model, noise is primarily caused by heat generated in electronic signal processing, leading to

performance degradation. We consider both shot noise and thermal noise during optical signal conversion at the PD [29]. Over a 10–100 km range, both DN and UP transmission signals are able to receive 0 dBm to −17 dBm of power, as shown in Figure 7a,b. In addition, 187.7 THz (in DN) and 195.5 THz (in UP) signals both decline sharply compared to other signals. Since the wavelengths of high-frequency signals (187.8 THz and 195.6 THz) are shorter than those of lower frequencies, they can travel farther without being absorbed or scattered by the atmosphere. Moreover, the lower-frequency signals travel less than the higher-frequency signals do over long distances due to the fact that the low-frequency signals experience fewer cycles than do higher-frequency signals. The 187.7 THz and 195.5 THz signals transmitted also suffer from higher inter-channel interference (via frequency offset) and inter-symbol interference (via multi-path channels).

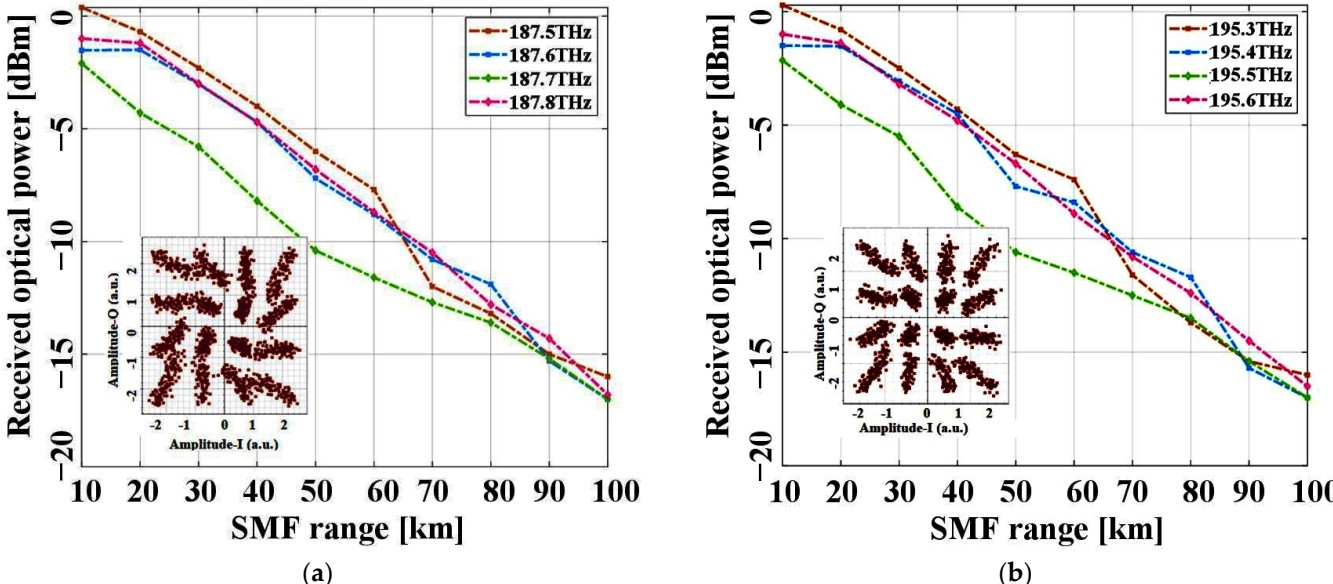

**Figure 7.** SMF range versus received optical power performance of different frequency signals in (**a**) DN and (**b**) UP directions; insets: corresponding constellation diagrams.

The optical power received for 187.5, 187.6, 187.7, and 187.8 THz signals is −16, −17, −17, and −16.9 dBm, for BER limits of 10-3 over 100 km fiber. As well, at a 100 km SMF range, 195.3, 195.4, 195.5, and 195.6 THz signals have −16, −17, −17, and −16.5 dBm power sensitivities. By increasing the transmitted power, the signal quality can be improved. Figure 7a,b show constellation diagrams for a 16-QAM modulation format on 30 and 80 km fiber at 187.7 THz in DN and 195.5 THz in UP. The BER is clearly below the limit over long-range transmissions for the constellations. Thus, the hybrid fiber–LiFi link enables transmission over long distances at aggregate data rates of 200 Gbps without requiring any digital processing units.

Table 2 illustrates the optical spectra of hybrid HS-PON/OFDM using the LiFi system for different fiber lengths and a fixed LiFi range of 10 m at an input power of −6 dBm and a 50 Gbps channel rate. The system utilizes four HS-PON downlink channels (187.5–187.8 THz) separated by a 100 GHz channel spacing, while the LiFi system uses one blue LED (450 nm or 666.2 THz).

With increasing SMF ranges from 10 to 100 km, the power in the main tones decreases, but increases in the side lobes. It is possible to obtain a maximum optical power of −30 dBm in a 10 km SMF range with an LED signal power of −62 dBm. In contrast, for SMF and LiFi channels at 100 km, a maximum optical power of −45 and −70 dBm, respectively, isobtained. In addition, the side lobes are between −60 dBm and −70 dBm over a 10–100 km fiber range. Owing to the essence of fiber impairments like linear and

non-linear effects, an optimum launch power at the transmitter in the model is able to minimize these effects.

**Table 2.** Optical spectra of the proposed model at different fiber lengths.

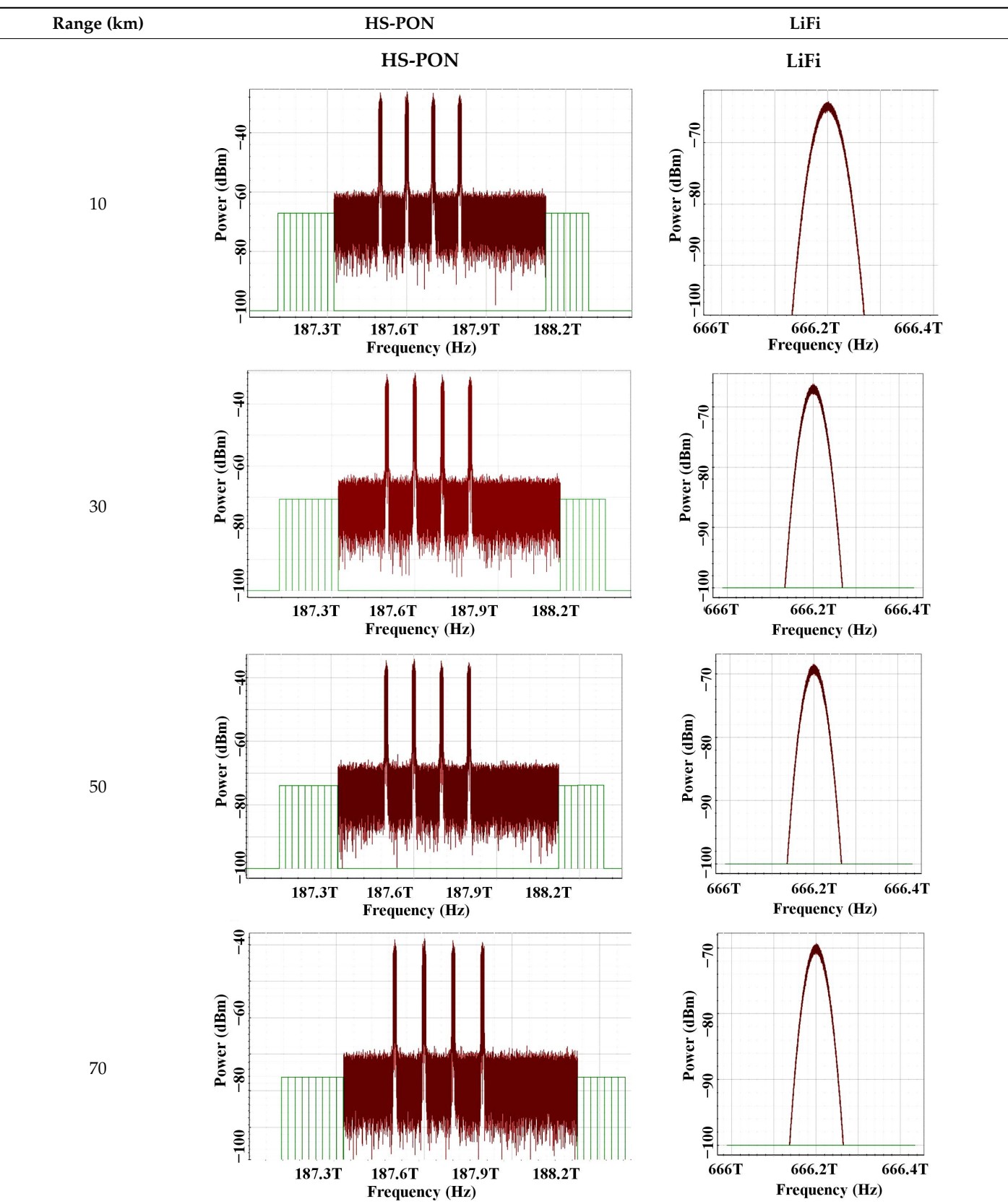

**Table 2.** *Cont.*

| Range (km) | HS-PON | LiFi |
| --- | --- | --- |
| 100 | | |

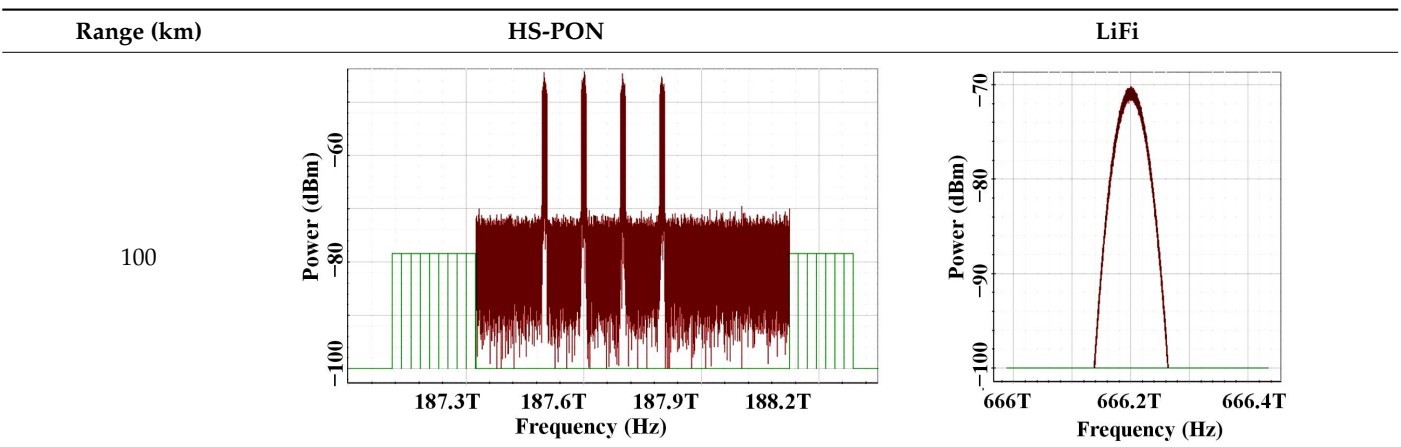

Compared to the recent multiple channels of OFDM-based PONs in Refs. [31–33] with a channel capacity of 2 × 4.2 Gbps [31], 3 × 1 Gbps [32], and 2 × 5.2 Gbps [33], the proposed HP-PONs/LiFi system exploits a significant amount of optical beams to offer greater capacity as well as more flexibility for end users at a symmetric and bidirectional data rate of 4 × 50 Gbps. This approach has acquired considerable attention from industry and academia [34].

Figure 8a,b illustrate the model's BER performance versus its throughput for DN and UP transmission over LiFi channels, respectively. Signals transmitted uplink perform slightly better than signals transmitted downlink. A maximum transmission data rate of 50 Gbps can be achieved at a BER limit of $10^{-3}$. Additionally, the BER values increase as the channel throughput increases. The minimum BER for DN and UP transmissions is $10^{-20}$ (187.5 THz) and $10^{-22}$ (195.3 THz), respectively.

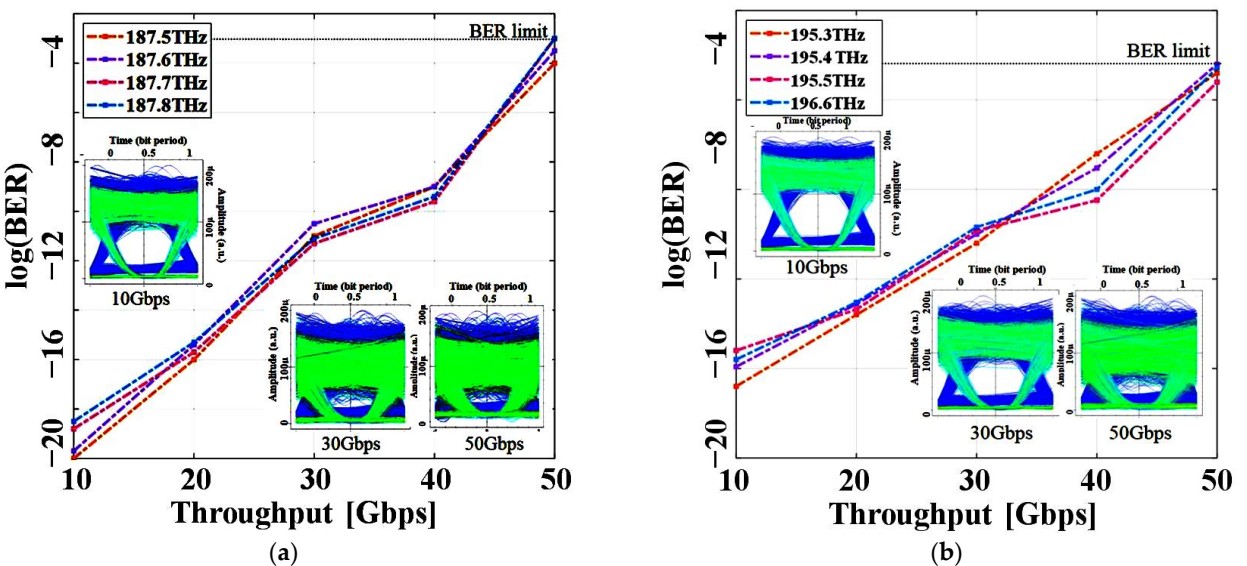

**Figure 8.** Throughput versus BER performance of different frequency signals in (**a**) DN and (**b**) UP transmission; insets: corresponding eye patterns.

In Figure 8a,b, the eye patterns for different DN and UP transmitted signals are more open at low throughputs, while at high throughputs of 50 Gbps, they are acceptable. This proves the feasibility of the proposed model at a high throughput. DN transmission signals, however, become more closed from 10 Gbps to 50 Gbps per channel, compared with UP

transmission signals. The most prominent degradation factor of hybrid PON/LiFi systems is the fiber inter-symbol interference and LiFi channel disruptions. Hybrid fiber–LiFi link technology is therefore promising because it can significantly enhance reliability and availability compared to individual channels, as well as providing adaptive solutions for high-throughput LiFi connectivity, transmission rates, and insensitivity to channel impairments [35]. Figure 9a,b present the achieved BER for different irradiance and incidence half-angle values for DN and UP transmission signals, respectively.

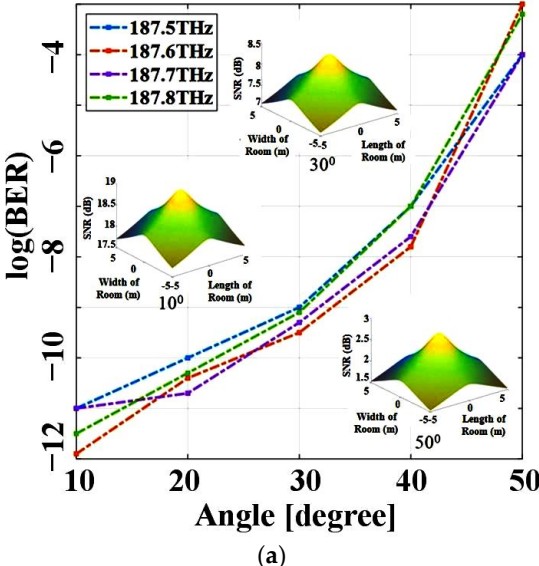
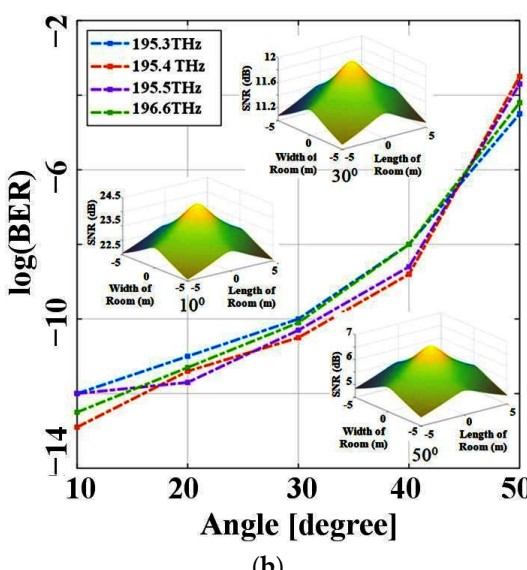

(a)  (b)

**Figure 9.** Tx and Rx angle versus BER performance of different frequency signals in (**a**) DN and (**b**) UP transmission; insets: corresponding Lambertian patterns.

The transmitter half-angle is set at 60°, while both the irradiance half-angle and incidence half-angle vary equally from 10° to 50°. In the case of indoor wireless communication via a LiFi medium, the performance of all transmission signals degrades as the angle increases. For downlink and uplink transmission, the lowest BER can be obtained as $10^{-12}$ (187.6 THz) and $10^{-13}$ (195.4 THz), at a 10° irradiance half-angle and 10° incidence half-angle, respectively. The logarithmic BER value increases from −12 to −3 for DN and −13 to −3 for UP transmission signals as angles increase from 10° to 50°.

Figure 9 also indicates the LiFi illuminance distribution in a 10 × 10 × 10 m sized room. Illuminance levels are highest in the center of the room, while they are significantly lower in the corners. A number of factors influence the illumination of an LED, including its location, field of view, orientation, output power, emission patterns, color temperature, bandwidth, etc. [36]. According to the results, the SNR in a directed path decreases with an increasing irradiance and incidence angle. As a result, Lambertian patterns have limited performance but large coverage areas. However, at an angle of angle, the SNR value is high, which means improved system performance in terms of an extended communication range and limited coverage area.

Figure 10a,b illustrate the BER performance of DN and UP transmission signals over a 10 km fiber and 10 m wireless range for the varied detection of surface areas of LiFi channels. Based on these plots, we can conclude that the detection surface area is still highly dependent on the system's performance. Multiple transmission signals can be received with greater power due to the increased surface area of LOS communication.

Figure 10 shows that the logarithmic BER value decreases down to −20 whenever the detection area exceeds 1.5 cm². As shown in Figure 10a, even at high data rates of 50 Gbps, the BER falls within the range of approximately $10^{-9}$ to $10^{-20}$ when the surface area ranges between 1.5 and 4 cm². Similarly, for UP transmission signals, the BER ranges from $10^{-9}$ to $10^{-20}$, which offers better performance than other ranges do, as shown in

Figure 10b. Increasing the receiving surface area of LiFi communication systems improves BER performance, allowing more connections to be established.

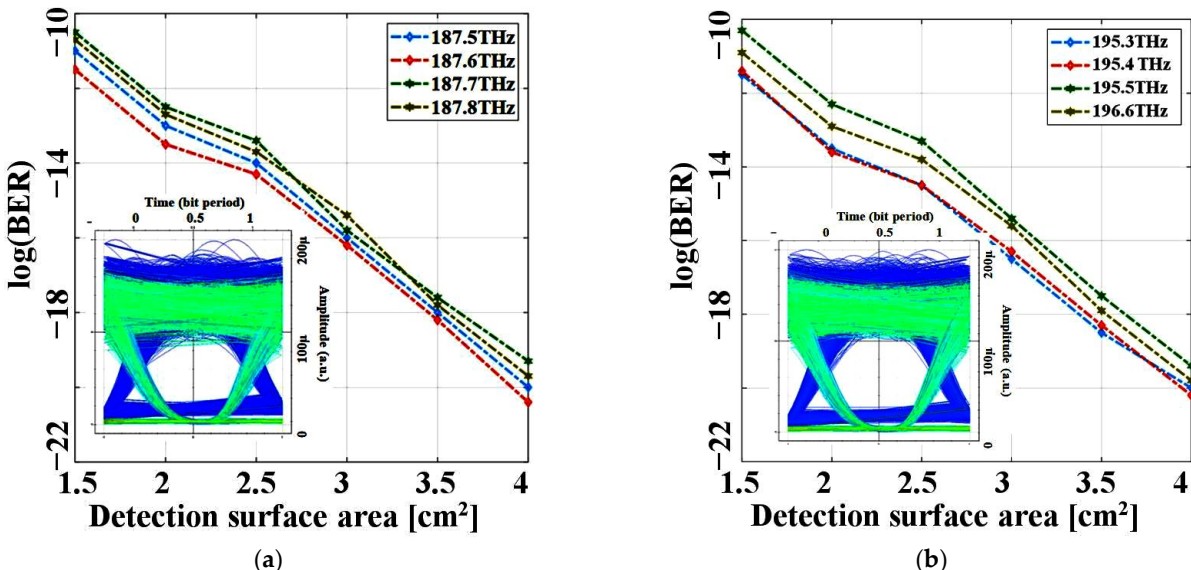

(**a**)                                                                (**b**)

**Figure 10.** Detection surface area versus BER performance of different frequency signals in (**a**) DN and (**b**) UP transmission; insets: corresponding eye patterns on 4 cm$^2$ surface area.

Also, for both DN and UP transmissions, on a 4 cm$^2$ detection surface area, a fine eye opening is observed. For LiFi systems with fixed transceivers, the Tx and Rx must be properly coordinated, which has several advantages [37].

The plots in Figure 11a,b show the BER performance versus the varied LiFi channel ranges with fixed 10 km fiber ranges in DN and UP directions, respectively. When LiFi channel ranges increase from 10 to 20 m, logarithmic BER values increase from −10 to −3 for DN and from −10.5 to −3 for UP transmission signals.

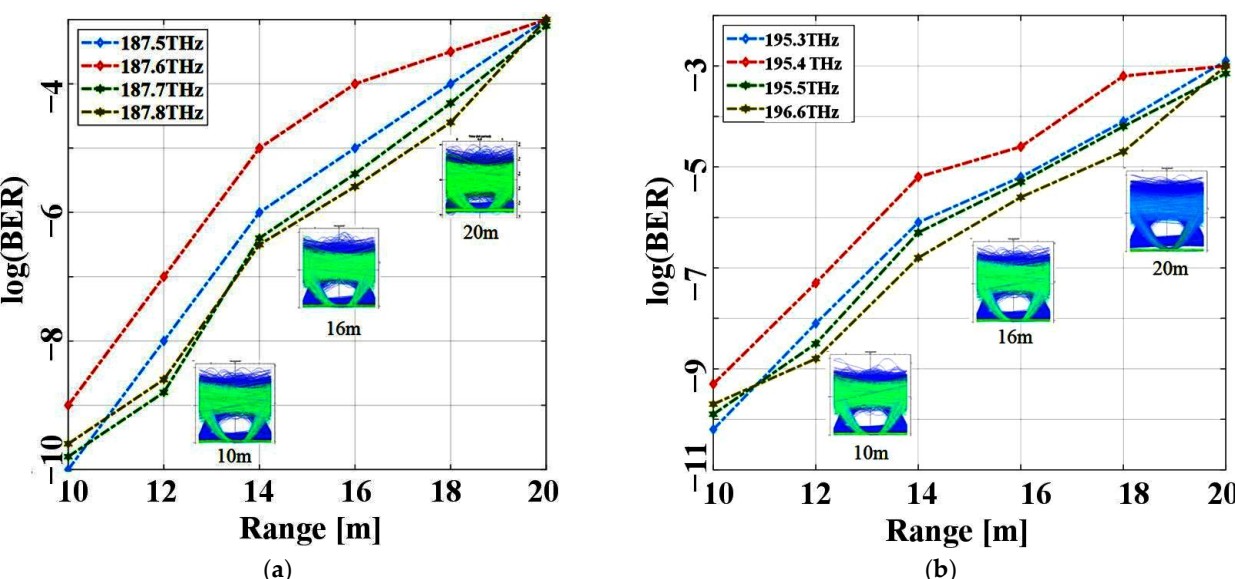

(**a**)                                                                (**b**)

**Figure 11.** LiFi range versus BER performance for (**a**) DN and (**b**) UP transmission; insets: corresponding eye patterns.

As the transmission range increases, the model BER performance increases for all transmission signals. This trend is due to the presence of LiFi channel impairments like channel noise and misalignments, which increases the disturbance as the range increases.

LiFi ranges of up to 20 m can be achieved using DN and UP transmission signals with BERs as low as $10^{-3}$, as illustrated in Figure 11a,b. Marginally better performance is also offered by UP transmission signals compared to DN transmission signals. Furthermore, it is observed that eye patterns at 10 m transmission ranges are wider and more open than those at 20 m transmission ranges in both directions.

For shorter transmission ranges, distinct DN and UP signals are observed as a clearer and wider eye opening than that for longer transmission ranges because the signals become less distorted by the generated noise. When signals are transmitted over a distance of 20 m, the eye pattern becomes thicker, indicating distorted signal quality.

Table 3 indicates the 3D and 2D BER patterns of the proposed model over various LiFi channel ranges with a fixed 10 km fiber at a 50 Gbps data rate. These patterns revealed that continuous and regular BER patterns are obtained ina 10 m range followed by a 14 and 18 m range. This is due to the presence of LiFi channel noise, link misalignments, and external light effects.

**Table 3.** Three-dimensional and two-dimensional view of 3D BER patterns of the proposed model.

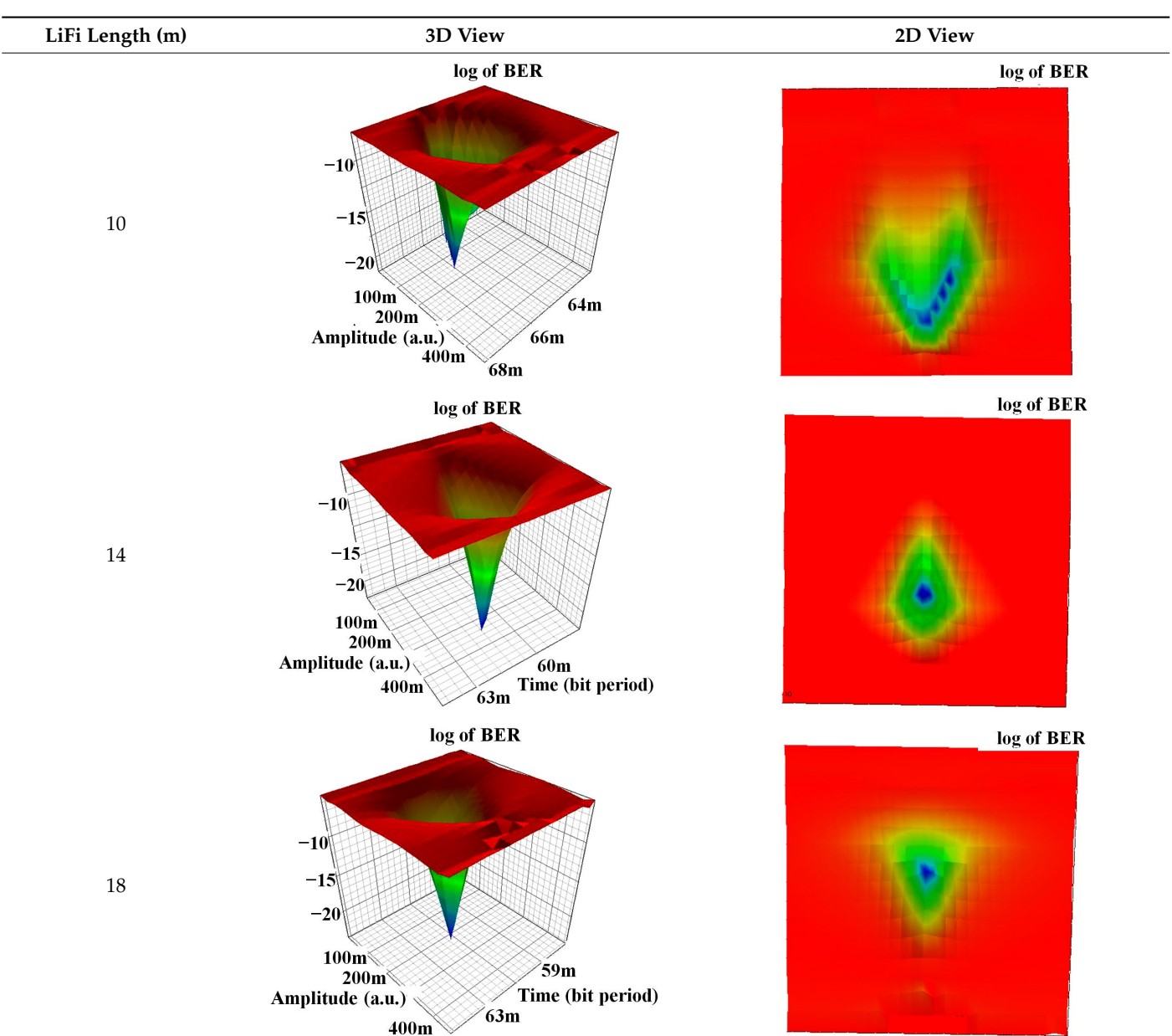

| LiFi Length (m) | 3D View | 2D View |
|---|---|---|
| 10 | | |
| 14 | | |
| 18 | | |

Table 4 indicates the performance evaluation of the proposed model over a 10 km SMF and a 10 m LiFi range at a 50 Gbps data rate.

**Table 4.** Performance evaluation of the model.

| Frequency (THz) | Gain (dB) | Noise Figure (dB) | Output SNR (dB) | Output OSNR (dB) |
|---|---|---|---|---|
| 187.5 | 20.03 | 3.29 | 7.76 | 16.79 |
| 187.6 | 20.01 | 3.26 | 7.73 | 16.74 |
| 187.7 | 20 | 3.28 | 7.75 | 16.76 |
| 187.8 | 20.05 | 3.24 | 7.76 | 16.79 |

The evaluated performance metrics include the gain, noise figure, SNR, and optical SNR, as given in expressions (20)–(22).

Table 5 illustrates the comparative analysis of existing methods. It shows that a high-speed full-duplex hybrid fiber–LiFi in a hybrid PON/LiFi model provides superior performance for multi-users.

**Table 5.** Comparison analysis of proposed model with prior works.

| Ref. | Platform | Technique | Max. Wireless Range (m) | Max. Fiber Distance (km) | Highest Data Rate (Gbps) | LED/ LD | Wavelength (nm) |
|---|---|---|---|---|---|---|---|
| [31] | Experiment | OFDM using hybrid laser diode (LD)–LED | Not defined | Not used | 4.2 | LD, LED | 785, 808 |
| [32] | Experiment | LED array | 3 | Not used | <1 | LED | 460, 525, 625 |
| [33] | Experiment | Multiple-input–multiple-output with spatial diversity, spatial multiplexing, and OFDM | 1 | Not used | 5.2 | LD | 520, 650 |
| [38] | Experiment | VLC | 0.15 | Not used | 1.25 | LD | 410 |
| [39] | Experiment | Wavelength division multiplexing OFDM | 1.6 | Not used | 15.73 | LED | 605, 567, 490, |
| [40] | Simulation (OptiSystem) | OCDMA-OFDM PON | Not used | 142 | 40 | Not used | - |
| [41] | Simulation (OptiSystem) | WDM-FSO | Not used | 30 | 10 | Not used | - |
| [42] | Simulation (OptiSystem) | DWDM | Not used | 50 | 40 | Not used | - |
| [43] | Simulation (OptiSystem) | OFDM-PON WITH FSO | Not used | 20 | 10 | Not used | - |
| Proposed work | Simulation (OptiSystem) | Bidirectional HS-PON/OFDM with LiFi system | 20 | 100 | 50 | LED | 450 |

As shown in the table, the prior work used three LEDs over 1.6 m to achieve a 15.73 Gbps data rate [39]. Previous work did not include fiber links. Compared to previous work, our proposed solution uses blue LEDs to offer 20 m LiFi communication at a rate of 4 × 50 Gbps on a 100 km fiber. Based on this, it becomes evident that the proposed model provides a high data rate, high bandwidth, scalability, flexibility, reliability, and easy upgradeability to next-generation optical networks. It is crucial to consider the trade-offs

between the performance of the designed model and power dissipation/complexity in a hybrid high-speed/LiFi network application that transmits coherent technology. A four-wavelength HS-PON with a single LED LiFi system is incorporated with a 16-QAM OFDM scheme to achieve a better tradeoff between computation complexity and communication performance for 5G and beyond-5G networks [44–46].

## 4. Conclusions

We propose a symmetric and bidirectional HS-PON based on LiFi using four 50 Gbps channels. A hybrid fiber–LiFi link transmits four 16-QAM OFDM-modulated signals in the downlink and four in the uplink. Results show that a hybrid HP-PON/OFDM-based LiFi system can offer a maximum fiber range of 100 km at a $-17$ dBm receiver sensitivity. Also, it enables indoor LiFi communication with a range of up to 20 m over a fixed 10 km fiber. When fiber–LiFi channel impairments and noise are taken into account, a maximum irradiance and incidence half-angle of 500 can provide reliable transmission at BER of $10^{-3}$. In addition, the system offers BERs between $10^{-9}$ and $10^{-20}$ over LiFi communication links with detection areas between 1.5 and 4 cm$^2$. Successfully obtaining a high SNR of 7.76 dB, high SNR of 20.05 dB, and high OSNR of 16.79 dB can be achieved with a gain of 20.05 dB, a noise figure of 3.24 dB, and a noise factor of 3.24 dB. In addition, a comparative analysis indicates that our proposed work outperforms existing ones in terms of its high transmission rate, long-haul hybrid fiber–LiFi communication, and cost-effectiveness.

This field offers numerous opportunities for future research. To address the security issues in the outdoor environment, pulse amplitude modulation is one of the modulation techniques that has been developed to support multi-access users and devices [42].

**Author Contributions:** M.K. and S.K.M. discussed the plan and agreed on it. M.K., S.K.M., M.B. and V.A. drafted designs for the manuscript. The original manuscript was written by M.K. and V.A. The manuscript was edited by M.B. and S.K.M. They all reviewed and commented on the original draft of the manuscript. All authors have read and agreed to the published version of the manuscript.

**Funding:** There is no funding involved for this research. To conduct this research, Satyendra K. Mishra would like to acknowledge CTTC Barcelona Spain.

**Institutional Review Board Statement:** Not applicable.

**Informed Consent Statement:** Not applicable.

**Data Availability Statement:** Data are contained within the article.

**Conflicts of Interest:** The authors declare no conflict of interest.

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
