# Peer review of "Investigation of OFDM-Based HS-PON Using Front-End LiFiSystem for 5G Networks"

_photonics, doi:10.3390/photonics10121384_

Round 1

Reviewer 1 Report

Comments and Suggestions for Authors

The paper has a lot of wording or spacing issues, such as in line 41, 42, 46, ... It made the reading very chanllege. Authors need to fix them before resubmitting.

Figure 5: Authors mentioned that 187.7 THz (in DN) and 195.5THz (in UP) 235 signals decline sharply than other signals, but no explaination of the reason. 

Line 260: No comparison data with other recent methods.

What is the tradeoff of the proposed work?

Author Response

Attached the documents

Reviewer 2 Report

Comments and Suggestions for Authors

Authors

The authors propose bidirectional OFDM based 4×50Gbps HS-PON architecture based on using LiFi system and 16-QAM modulation. The proposed system can achieve fiber range of 100 km and enable indoor front-end LiFi communication over a range  up to 20 m with fixed 10 km fiber length. Iti s aimed to meet the requirements for the 5G front end communication networks. The presented results  show the advantage of the proposed system compared with the conventional soultions. The manuscript contains some interesting research results. The presentation should be improved.

There are many typos in the text. It has to be checked and corrected.

The quality of figures should be improved. The letters on the figures should be readable.

Comments on the Quality of English Language

Author Response

Attached the documents

Reviewer 3 Report

Comments and Suggestions for Authors

In the peer-reviewed manuscript, the authors analyse an OFDM-based symmetric and bidirectional 4×50Gbps high-speed passive optical network (HS-PON) using light fidelity (LiFi) system.

Experimental results show that the proposed hybrid HP-PON/OFDM using blue LED-based LiFi system can offer a maximum fibre reach of 100km with a receiver sensitivity of -17dBm. In addition, it enables indoor front-end LiFi communication up to 20m with a fixed fibre length of 10km.

The paper brings original novel information in the domain of the journal’s thematic focus. The research results are clearly distinguished from results adopted and used literary resources are mentioned properly. Credibility of published results is documented (experiments - simulations). Text readability and its linguistic correctness (even English texts, especially in the case of the technical terminology) is on the appropriate level.

I have the following comments on the scientific content of the article and I ask the authors to comment on them:

1. A bit error rate (BER) better than 10e-5 is considered acceptable in wireless LAN applications. You mention for your LiFi system BER 10e-3 !!! is it sufficient???

2. A BER better than 10e-9 is considered acceptable in passive optical network (PON) applications. You mention bit error rate up to 10e-20!!! is it realistic to achieve???

I have the following comments on the formal side of the article:

1. the keywords need to be arranged alphabetically,

2. the abstract needs to be edited: clearly distinguish what is the goal and what is the benefit of the work!!!,

3. the Table 2 shows the dependence of Optical Power on Frequency for HS-PON and, at the same time, for LiFi Optical Power on Wavelength. For simplicity, clarity and faster orientation, the x-axis should also be the same (either frequency or wavelength).

In general, after appropriate corrections and additions, I approve publication of this manuscript.

Author Response

Attached the document

Reviewer 4 Report

Comments and Suggestions for Authors

Review of manuscript Photonics-2714110

Investigation of OFDM based HS-PON using front-end LiFi system for 5G networks

By Meet Kumari, Mai Banawan, Vivek Arya and Satyendra Kumar Mishra.

The authors proposed and investigated theoretically a novel hybrid high-speed  passive optical network (HS-PON) system using orthogonal frequency division multiplexing (OFDM) and light fidelity (LiFi) for the 5G applications such as mobile, cloud computing and fiber-wireless convergence. They analyzed the performance of the proposed system. They carried out the numerical simulations and evaluated the bit error rate (BER), receiver sensitivity, and optical signal to noise ratio (OSNR). The results are presented in a number of tables and figures. The manuscript is novel and can be interesting for the researchers and engineers occupied in modern optical communication systems. However, the manuscript cannot be accepted for publication in a present form. It requires the following major revisions.

 The editing of the text is necessary because of many misprints and errors. See for instance the following misprints.

Page 1, line 33. “…enablingtechnologies…”.

Page 1, lines 38-39. “…entertainment.most of seaapplications…”

Page 1, line 39. “…users.Due to the…”.

Page 2, line 46. “…efficient.LiFi…”.

Page 2, line 52. “…pureLiFi…”.

Page 2, line 70. “…and cost-effectivetransceivers[11].”

Page 3, line 100. “…systemin…”.

Page 3, line 116. “In indoor LiFiscenarios…”

Page 15, line 335. “…a clearer andwider eye…’

Page 18, line 378. ‘…there arenumerous opportunities…”

In Section 2 it would be helpful to insert the mathematical description of the optical signals propagating in the proposed system.

 I recommend a separation of Figures 2 a, b, c into three separate figures with more detailed captions.

The quality of Figures 3(a0-3(p) is low. I recommend enlarging each figure in order to make numbers at the axes more pronounced.

The 2d BER patterns in Table 3 are not visible.

Comments on the Quality of English Language

There are many misprints and errors in the text. Some of them are presented in the Review.

Author Response

Attached the document

Round 2

Reviewer 1 Report

Comments and Suggestions for Authors

All comments are addressed by the revised version. 

Author Response

I appreciate your recommendation for publication.

Reviewer 3 Report

Comments and Suggestions for Authors

All my suggested comments have been taken care.

In general, I agree to the publication of this manuscript.

Author Response

I appreciate your recommendation for publications.

Reviewer 4 Report

Comments and Suggestions for Authors

Review of the revised manuscript Photonics-2714110

Investigation of OFDM based HS-PON using front-end LiFi system for 5G networks

By Meet Kumari, Mai Banawan, Vivek Arya and Satyendra Kumar Mishra.

The authors have taken into account the comments and recommendations of the first review and made the necessary revisions. They corrected the misprints and errors in the text of the manuscript.

They introduced the new Subsection 2.1 containing the mathematical description of the model.

Figure 2 is separated into Figures 2,3, and 4 with the corresponding captions.

The authors introduced a new Figure 5 Simulation diagram of the proposed model in OptiSystem software.

The quality of Figures 6 (a)-(p) and the 2d BER patterns in Table 3 is improved.

The manuscript in general is substantially improved by the authors. The revised manuscript can be accepted for the publication in a present form.

Comments on the Quality of English Language

Author Response

(The authors gave the same response as above.)
